# Synthesis, Characterization, and Bactericidal Activity of a 4-Ammoniumbuthylstyrene-Based Random Copolymer

**DOI:** 10.3390/polym13071140

**Published:** 2021-04-02

**Authors:** Silvana Alfei, Gabriella Piatti, Debora Caviglia, Anna Maria Schito

**Affiliations:** 1Department of Pharmacy, University of Genoa, Viale Cembrano, 16148 Genoa, Italy; 2Department of Surgical Sciences and Integrated Diagnostics (DISC), University of Genoa, Viale Benedetto XV, 6, 16132 Genova, Italy; gabriella.piatti@unige.it (G.P.); Caviglia86@gmail.com (D.C.); amschito@unige.it (A.M.S.)

**Keywords:** multi-drug-resistant Gram-positive and Gram-negative clinical isolates, cationic antibacterial copolymers, 4-ammoniumbuthylstyrene hydrochloride, membrane disruptors, MIC and MBC values, time-killing experiments, turbidimetric studies

## Abstract

The growing resistance of bacteria to current chemotherapy is a global concern that urgently requires new and effective antimicrobial agents, aimed at curing untreatable infection, reducing unacceptable healthcare costs and human mortality. Cationic polymers, that mimic antimicrobial cationic peptides, represent promising broad-spectrum agents, being less susceptible to develop resistance than low molecular weight antibiotics. We, thus, designed, and herein report, the synthesis and physicochemical characterization of a water-soluble cationic copolymer (P5), obtained by copolymerizing the laboratory-made monomer 4-ammoniumbuthylstyrene hydrochloride with di-methyl-acrylamide as uncharged diluent. The antibacterial activity of P5 was assessed against several multi-drug-resistant clinical isolates of both Gram-positive and Gram-negative species. Except for strains characterized by modifications of the membrane charge, most of the tested isolates were sensible to the new molecule. P5 showed remarkable antibacterial activity against several isolates of genera *Enterococcus*, *Staphylococcus*, *Pseudomonas*, *Klebsiella*, and against *Escherichia coli*, *Acinetobacter baumannii* and *Stenotrophomonas maltophilia*, displaying a minimum MIC value of 3.15 µM. In time-killing and turbidimetric studies, P5 displayed a rapid non-lytic bactericidal activity. Due to its water-solubility and wide bactericidal spectrum, P5 could represent a promising novel agent capable of overcoming severe infections sustained by bacteria resistant the presently available antibiotics.

## 1. Introduction

Cationic macromolecules including cationic dendrimers (CDs) [1], cationic polymers (CPs) and positively charged amphiphilic copolymers (CACs) [2] increasingly attract the attention of scientists and of various sectors of industry, such as food and food packaging, due to their capability to limit or inhibit bacterial growth, both in solution and on surfaces. CDs, CPs, and CACs are active because they are positively charged macromolecules that, mimicking natural cationic antimicrobial peptides (CAPs), can kill bacteria on contact, causing irreparable damage in the anionic cell membranes up to their disruption [1,2]. Monomers containing permanently cationic tetra alkyl ammonium groups have been widely employed to prepare homo-polymers and copolymers, capable of interacting with negative constituents of bacterial surface of Gram-positive and Gram-negative pathogens, and to reach the cytoplasmic membranes by diffusion through the outer membrane (Gram-negative) and the cell walls (Gram-negative and Gram-positive) [1,2,3,4,5,6]. Due to these electrostatic interactions and diffusion processes, membranes depolarize and progressive permeabilization occur, leading to membrane disruption, loss of cytoplasmic content and bacterial death [1,2,3,4,5,6].

The search for new cationic antibacterial polymers, more selective for bacteria cells, led to a new trend, that favors using monomers with primary ammonium groups in the form of acidic salts, to prepare cationic macromolecules capable of mimicing the amphiphilic properties and cationic functionalities of CAPs containing lysine [2,3,4,5,6,7]. Comparisons between compounds containing primary, tertiary, and quaternary ammonium groups, revealed that the polymers containing primary ammonium groups outperformed their tertiary and quaternary analogs in terms of higher antibacterial activity and lower toxicity [2,7]. The preparation of random copolymers with uncharged acrylates, methacrylates, acrylamides and methacrylamides, with the aim of reducing the cationic charges, allowed to tune the balance between hydrophobic and cationic properties, as well as to control the length of the polymers [7].

The advantages associated with the use of CPs or CACs, acting (as CAPs) mainly in virtue of a non-specific electrostatic interaction with the negatively charged bacterial membranes, include a fast inhibitory effect, a high selectivity for pathogens and a limited tendency to develop resistance. This latter represents one of the most alarming global problems, which lead to untreatable infections, huge health care costs and high mortality rates [1,2,5,6,8].

The selectivity of CPs and CACs towards bacterial membranes depends on fundamental differences in the lipid composition of the cell membrane and in the surface’s components that exist between the prokaryotic and eukaryotic cells [8,9]. 

Typically, the cytoplasmic membrane of mammalian cells is less anionic than that of bacteria, in terms of charge. For example, the human erythrocyte membrane is mainly composed of neutral lipids, such as phosphatidylcholine, sphingomyelin, and phosphatidylethanolamine, with a minor fraction of negatively charged phosphatidylserine [8,10]. On the contrary, the microbial cell membrane is composed of more negatively charged molecules such as phosphatidylserine, phosphatidylglycerol or cardiolipin, and more easily attracts cationic molecules, such as CDs, CPs, and CACs. In Gram-positive bacteria, teichoic acids, which are linked to either the peptidoglycan cell wall or to the underlying cell membrane, impart to the cell surface net negative charges. As for Gram-negative bacteria, they have an additional outer membrane, containing phospholipids and lipopolysaccharides, which impart a strongly negative charge to the cell surface [8]. Moreover, in mammalian cells, the presence of different classes of sterols or similar molecules, absent in prokaryotes, confers rigidity to the cytoplasmic membrane. Therefore, they carry out a significant protective action against the damaging action of cationic macromolecules, which direct their action preferentially towards bacteria membranes [8,11].

The increasing prevalence of antibiotic-resistant bacterial infections, together with the considerable scarcity of new curative drugs, render molecules, such as synthetic CDs and CACs (which have shown broad antimicrobial spectrum, rapid bactericidal kinetics, and a very low propensity to induce resistance), of great interest to the scientific community [1,2,7]. 

Antibiotic resistance is easily acquired through several unrelated mechanisms that include, among others, mutation of targets, the action of specific enzymes and modification of cell permeability.

On the contrary, polycationic molecules, which act in a non-specific way upon simple contact with cell surfaces, do not need to enter bacteria and interact with specific and mutable enzymatic processes [1,2]. Consequently, it is extremely unlikely that pathogens will be able to develop resistance against these macromolecules active directly on the cell membrane [8].

Concerning the synthetic procedure, the so-called random copolymers can be basically synthesized by combining a non-polar monomer with a cationic monomer. These structures have both positive charges and neutral moieties, randomly situated along the polymer backbone, thus, better modulating the cationic character, which is fundamental for antibacterial activity and resulting less hemolytic.

Within this context, we herein reported the design of the 4-ammoniumbuthylstyrene hydrochloride monomer (M5) and its radical copolymerization with the uncharged di-methyl acrylamide (DMAA) as a comonomer. A highly hydrophilic water-soluble cationic copolymer (P5) was obtained, which showed to possess physicochemical properties, in terms of particle size, Z-potential (ζ-p), solubility and stability in water solution, suitable for a possible clinical application. The antibacterial activity of P5 was assessed against many multidrug resistant (MDR) clinical isolates of Gram-positive and Gram-negative species, through MIC and MBC determinations. Furthermore, time-killing experiments and turbidimetric studies were performed to confirm the biocidal activity and to investigate the mechanism of action of P5. 

## 2. Materials and Methods 

### 2.1. Chemicals and Instruments

Monomer M5 (**5**) was prepared following a procedure previously reported [12], but herein, opportunely modified. The procedure performed in this work has been described in detail in Section 2.2. Melting points and boiling points are uncorrected. FTIR spectra were recorded as films or KBr pellets on a Perkin Elmer System 2000 instrument (PerkinElmer, Inc., Waltham, MA, USA). ^1^H and ^13^C NMR spectra were acquired on a Bruker DPX spectrometer (Bruker Italia S.r.l., Milan, Italy) at 300, and 75.5 MHz, respectively. Fully decoupled ^13^C NMR spectra were reported. Chemical shifts were reported in ppm (parts per million) units relative to the internal standard tetramethylsilane (TMS = 0.00 ppm), and the splitting patterns were described as follows: s (singlet), d (doublet), t (triplet), q (quartet), m (multiplet), and br (broad signal). Mass spectra were obtained with a GC-MS Ion Trap Varian Saturn 2000 instrument (Varian, Inc., Palo Alto, CA, USA; EI or CI mode; filament current: 10 mA) equipped with a DB-5MS (J&W) capillary column. Elemental analyses were performed with an EA1110 Elemental Analyser (Fison Instruments Ltd., Farnborough, Hampshire, England).

High Performance Liquid Chromatography (HPLC) analyses were performed on a Jasco model PU-980 instrument (JASCO Corporation, Hachioji, Tokio, Japan), equiped with a Jasco Model UV-970/975 intelligent UV/Vis detector (JASCO Corporation, Hachioji, Tokio, Japan) at room temperature. A constant flow rate (1 mL/min), UV detection at 254 nm, a Hypersil ODS 5 mm column and a mixture acetonitrile/water 6/4 as eluent, were employed for the acquisitions. GC-FID analyses were performed on Perkin Elmer Autosystem (Varian, Inc., Palo Alto, CA, USA), using a DB-5, 30 m, diameter 0.32 mm, film 1 mm capillary column. Column chromatography were performed on Merck silica gel (70–230 mesh). Dynamic Light Scattering (DLS) and Z-potential determinations were performed on the same instrument and with the same modalities previously described [13]. A thin layer chromatography (TLC) system employed aluminium-backed silica gel plates (Merck DC-Alufolien Kieselgel 60 F254, Merck, Washington, DC, USA), and detection of spots was made by UV light (254 nm), using a Handheld UV Lamp, LW/SW, 6W, UVGL-58 (Science Company^®^, Lakewood, CO, USA). The molecular weight of copolymer was determined on a vapor pressure Knauer K-700 osmometer (Advanced Scientific Instruments Wissenschaftliche Gerätebau, Berlin, Germany) in MeOH at 45 °C. 

All reagents and solvents were purchased from Merck (formerly Sigma-Aldrich, Darmstadt, Germany) and were purified by standard procedures. 2-2′-azobisisobutirronitrile (AIBN) was crystallized from methanol. Organic solutions were dried over anhydrous magnesium sulphate and were evaporated using a rotatory evaporator operating at a reduced pressure of about 10–20 mmHg.

### 2.2. Synthesis of 4-(4-Amminobuthyl)styrene Hydrochloride M5 (***5***)

#### 2.2.1. *p*-Vinylphenylmagnesium Chloride (**1**)

Magnesium (0.9680 g, 39.8 mmol), which was previously washed with anhydrous diethyl ether (Et_2_O) (10 mL), anhydrous tetrahydrofuran (THF) (5 mL) and a few goggles of methyl iodide (MeI) and 4-chlorostyrene were introduced into a 100 mL three-necked flask flamed under nitrogen equipped with mechanical stirrer, condenser, dropping funnel. After starting the reaction by overheating with flame, the mixture, kept under reflux, was added with a solution of 4-chlorostyrene (5.20 g, 37.5 mmol, 4.5 mL) in THF_dry_ (50 mL) by slow dripping. After further reflux by stirring and disappearance of the metallic magnesium (90′), stirring was continued at r.t. for 1 h. Then, the suspension was decanted to obtain a clear solution of Grignard’s reagent (**1**), which will be used as it is in the subsequent reaction. To understand the exact title of reagent **1**, an aliquot of the obtained solution (1 mL) was transferred in a flask containing 0.1014 N HCl in excess (10 mL) and back titrated with 0.1021 N NaOH (5.20 mL) in the presence of phenolphthalein as indicator. Reagent **1** resulted 0.580 N (85% yield).

#### 2.2.2. 4-(4-Bromobuthyl)styrene (**2**)

A mixture of the 1,4-dibromobuthane (24.72 g, 114.5 mmol), dry THF (50 mL) and a solution of LiCuBr_2_ in dry THF (3.4 mL) was cooled to 0 °C, treated dropwise with 0.580 N 4-vinylphenyl magnesium chloride (47 mL, 27.3 mmol) in THF and stirred at room temperature for 18 h. The reaction mixture was then treated with an iced aqueous solution of NaCN (1.81 g) and NH_4_Cl (11.34 g) dissolved in water (70 mL) and extracted with peroxide-free ethyl ether (3 × 60 mL). The extracts were dried over anhydrous MgSO_4_ overnight and the solvent was removed by evaporation at reduced pressure obtaining a pale-yellow oil (23.72 g). The unreacted 1,4-dibromobuthane was removed by distillation at reduced pressure (0.05 torr) and the oily yellow residue was furtherly purified by chromatographic column (petroleum ether 40–60 °C/acetone = 85/15) to provide **2** as colorless oil (5.01 g, 21.0 mmoli, 77% yield).

Bp. 100 °C/0.15 torr, (lit. [14]: 92–93 °C/0.1 torr). FTIR (film, ν cm^−1^) 990, 906 (CH_2_=CH). ^1^H NMR (CDCl_3_, 300 MHz, δ ppm) 1.76 (m, 2H); 1.86 (m, 2H); 2.61 (t, 2H, *J* = 7.4 Hz); 3.39 (t, 2H, *J* = 6.6 Hz); 5.19 (dd, 1H, *J*_1_ = 1.0 Hz, *J_cis_* = 10.9 Hz); 5.70 (dd, 1H, *J*_1_ = 1.0 Hz, *J_trans_* = 17.6 Hz); 6.68 (dd, 1H, *J_cis_* = 10.9 Hz, *J_trans_* = 17.6 Hz); 7.11–7.34 (m, 4H). ^13^C NMR (75.5 MHz, δ ppm) 29.74; 32.18; 33.60; 34.66; 113.05; 126.24; 128.54; 135.36; 136.61; 141.50. GC-MS (EI, *m*/*z*, %): 240 (M^+^ [^81^Br], 37); 238 (M^+^ [^79^Br], 34); 117 (100).

#### 2.2.3. *N*-[(4-Vinylphenyl)buthyl]phthalimide (**3**)

A mixture of **2** (1.17 g, 4.9 mmol), potassium phthalimide (0.9367 g, 5.1 mmol) and dry *N*,*N*-dimethylformamide (DMF) (6 mL) was heated at 55 °C under nitrogen and mechanical stirring for two nights. The reagent disappearance was confirmed by TLC (toluene 100%). After removal of the solvent at reduced pressure the white solid residue was taken with chloroform (32 mL), filtered, and washed with chloroform (3 × 10 mL). All the organic extracts were combined, washed with 0.2137 N NaOH (10 mL, pH = 14), water (2 × 5 mL, pH = 7–8) and dried over anhydrous MgSO_4_. The removal of the solvent at reduced pressure afforded **3** as a crude solid (1.22 g) which was furtherly purified by chromatographic column using toluene and then toluene 100% as eluents obtaining a white solid (1.05 g, 3.5 mmoli, 71% yield). Mp. 114–115 °C. Purity 98% by HPLC. IR (KBr, ν cm^−1^) 1703 (C=O), 992, 913 (CH_2_=CH). ^1^H NMR (CDCl_3_, 300 MHz, δ ppm) 1.69 (m, 4H); 2.64 (t, 2H, *J* = 7.0 Hz); 3.71 (t, 2H, *J* = 7.0 Hz); 5.18 (dd, 1H, *J_gem_* = 1.0 Hz; *J_cis_* = 10.9 Hz); 5.69 (dd, 1H, *J_gem_* = 1.0 Hz; *J_trans_* = 17.6 Hz); 6.68 (dd, 1H, *J_cis_* = 10.9 Hz; *J_trans_* = 17.6 Hz); 7.13–7.36 (m, 4H); 7.67–7.85 (m, 4H). ^13^C NMR (75.5 MHz, δ ppm) 28.14, 28.53, 35.07, 37.76, 112.94, 123.18, 126.20, 128.59, 132.14, 133.87, 135.27, 136.66, 141.73, 168.43. GC-MS (EI, *m*/*z*, %): 305 (M^+^, 100). Anal. Calcd. For C_20_H_19_NO_2_: C, 78.66; H, 6.27; N, 4.59. Found: C, 78.62; H, 6.26; N, 4.58.

#### 2.2.4. 4-(4-Amminobuthyl)styrene (**4**)

*N*-[(4-Vinylphenyl)buthyl]phthalimide (**3**) (3.05 g, 10.0 mmol) was dissolved in 95% ethanol (18 mL) and treated under nitrogen and stirring at reflux with a solution of hydrazine hydrate (0.72 g, 14.4 mmol, 0.7 mL) in 95% ethanol (2 mL) for 5 h up to the disappearance of **3** (TLC, eluent toluene). After removal of the solvent at reduced pressure, the solid residue was taken with chloroform (50 mL) and treated with 20% aqueous NaOH (30 mL). The aqueous phase was separated, extracted with chloroform (5 × 30 mL) and the extracts combined and dried over MgSO_4_. The removal of chloroform afforded the free base as oil (1.73 g, 9.9 mmoli, 99% yield) which was transformed into its hydrochloride (following Section 2.2.5) without distillation. IR (KBr, ν cm^−1^) 3302 (NH), 992, 904 (CH_2_=CH). ^1^H NMR (CDCl_3_, 300 MHz, δ ppm) 1.16 (bs, 2H, NH_2_); 1.46 (m, 2H); 1.62 (m, 2H); 2.60 (t, 2H, *J* = 7.4 Hz); 2.68 (t, 2H, *J* = 7.0 Hz); 5.17 (dd, 1H, *J_gem_* = 1.0 Hz; *J_cis_* = 10.9 Hz); 5.69 (dd, 1H, *J_gem_* = 1.0 Hz; *J_trans_* = 17.6 Hz); 6.68 (dd, 1H, *J_cis_* = 10.9 Hz; *J_trans_* = 17.6 Hz); 7.10–7.32 (m, 4H).

#### 2.2.5. 4-(4-Amminobuthyl)styrene Hydrochloride (**5**)

A solution of amine **4** (1.73 g, 9.9 mmol) in dry diethyl ether (200 mL) was cooled to 0 °C and treated under stirring up to saturation with dry gaseous hydrochloric acid (30′). The white precipitate was filtered, washed with fresh ether (50 mL), dried to constant weight obtaining a white solid (2.21 g) which was crystallized twice from 200 mL of acetonitrile (ACN) to afford the hydrochloride derivative **5** (namely M5) as flaky white solid (1.15 g, 5.42 mmoli, 55% yield). Mp 198–200 °C (acetonitrile). IR (KBr, ν cm^−1^) 3435 (NH_3_^+^), 991 and 911 (CH_2_=CH). ^1^H NMR (CD_3_OD, 300 MHz, δ ppm) 1.63–1.75 (m, 4H); 2.57–2.69 (m, 2H); 2.90–2.95 (m, 2H); 5.16 (dd, 1H, *J_gem_* = 1.1 Hz; *J_cis_* = 10.9 Hz); 5.71 (dd, 1H, *J_gem_* = 1.1 Hz; *J_trans_* = 17.6 Hz); 6.69 (dd, 1H, *J_cis_* = 10.9 Hz; *J_trans_* = 17.6 Hz); 7.10–7.36 (m, 4H). ^13^C NMR (75.5 MHz, δ ppm) 28.08, 29.16, 35.89, 40.70, 113.26, 117.32, 129.70, 136.97, 137.99, 142.63. Anal. Calcd. for C_12_H_18_ClN: C, 68.07; H, 8.57; N, 6.62; Cl, 16.74. Found: C, 68.05; H, 8.58; N, 6.64; Cl, 16.78.

### 2.3. Preparation of Copolymer P5 by Radical Copolymerization in Solution

In a 25 mL tailed test tube equipped with a magnetic stirrer and carefully flamed under nitrogen, monomer M5 (**5**) DMAA, AIBN as radical initiator, and the freshly distilled anhydrous solvent were introduced in the ratios reported in (Table 1). The mixture thus obtained was subjected to three vacuum-nitrogen cycles to remove the oxygen. The clear solution was then siphoned into a 25 mL flask with screw cap and silicone septum. Nitrogen was then bubbled for 5 min in the solution, which was subsequently left under stirring at 60 °C. The final yellow solution was evaporated at reduced pressure achieving the crude copolymer which was subjected to three cycles of dissolution in MeOH and precipitation in Et_2_O obtaining P5 as white solid. P5 was subsequently subjected to fractioning. 

#### Fractioning of P5

A solution of P5 in just enough MeOH was filtered and transferred in a three-necks round-bottomed flask equipped with a mechanic stirrer and a funnel. It was thermostated at 25 °C and the clear solution (S1) was slowly added with Et_2_O until an oily precipitate (OP5-1) was obtained. OP5-1 was decanted and separated from the supernatant (S2).

S2 was treated as the starting solution (S1) obtaining a second oily precipitate (OP5-2). OP5-1 and OP5-2 were then dissolved in MeOH and precipitated in an excess of Et_2_O obtaining the corresponding copolymers, namely P5-High and P5-Low. 

FTIR (KBr, ν cm^−1^) 3500 (NH_3_^+^); 1649 (C=O); 755 (*o*-disubstituted phenyl ring).

The unreacted monomer M5 was recovered from the mixture of the combined solvents by evaporation at reduced pressure.

### 2.4. Determination of the Average Molecular Mass (Mn) of P5

#### 2.4.1. Calibration

Solutions of polyoxyethylene (PEO) with Mn 10800 in MeOH were prepared at three different concentrations [c (mol/Kg)] and were analyzed by the vapor pressure osmometer (VPO) technique at 45 °C. The quotients of measured values (MV) and the corresponding concentrations, i.e., MV/c (Kg/mol), were determined. These data (Table 2) were reported in graph, and a linear regression curve whose equation was Equation (1) was obtained by Least Squares Method. By extrapolating it to concentration c = 0, *K_cal_* was determined, which was found to be 501.
(1)y=73442x+500.92

#### 2.4.2. Measurements

Solutions of P5 in MeOH were prepared at three different concentrations c (g/Kg) (Table 2) and were analyzed by VPO method at 45 °C. The ratios between the measurement values (MV) and concentrations (c) (kg/g) were plotted vs. concentrations (c) finding a regression curve whose extrapolation to concentration c = 0 provided the *K_meas_* (kg/g) for P5 (Table 2). The average molecular mass (*Mn*) of P5 was determined with equation Equation (2) and was reported in Table 2 in Section 3.3: (2)$$Mn\;\left(\frac{g}{mol}\right)=\;\frac{Kcal}{Kmeas}.\;$$

### 2.5. Determination of NH_2_ Equivalents Contained in P5

The NH_2_ content of P5, in the form of hydrochloride, was obtained by volumetric titrations with a solution of HClO_4_ in acetic acid (AcOH), using quinaldine red as indicator [15]. Briefly, acetic anhydride (3 mL) was added to a solution of HClO_4_ 70% (1.4 mL) in AcOH (80 mL), obtaining a colorless solution that was left stirring at room temperature overnight. The clear yellow solution was made up to 100 mL with AcOH and standardized with potassium acid phthalate. The title of solution was found to be 0.1612 N. A sample of P5 (300.5 mg) was dissolved in AcOH (5 mL), treated with 2 mL of a solution of mercury acetate (1.5 g) in AcOH (25 mL), added with a few drops of a solution of quinaldine red (100 mg) in AcOH (25 mL) and titrated with the standardized solution of HClO_4_ in AcOH, using a calibrated burette with needle valve (0.02 mL). The very sharp end points were detected by observing the disappearance of the red color. Standardization and titrations were made in triplicate and the results were reported as means ± standard deviation (SD), and were expressed both as µequiv. NH_2_/µmol of P5 and µequiv. NH_2_/g of P5. 

### 2.6. Dynamic Light Scattering (DLS) Analysis

The hydrodynamic size (diameter) (Z-AVE, nm) and polydispersity index (PDI) of P5 particles were determined using Dynamic Light Scattering (DLS) analysis. Z-Ave and PDI measurements were performed in water mQ as medium at max concentration of P5 of 3 mg/mL (pH = 7.4), in batch mode using a low volume quartz cuvette (pathlength, 10 mm). The analysis was performed by a photon correlation spectroscopy (PCS) assembly, equipped with a 50 mW He-Ne laser (532 nm) and thermo-regulated at the physiological temperature of 37 °C. The scattering angle was fixed at 90°. Results were the combination of three 10-min runs for a total accumulation correlation function (ACF) time of 30 min. The hydrodynamic particle size result was volume-weighted and reported as the mean of three measurements ± SD). PDI value was reported as the mean of three measurements ± SD made by the instrument on the sample. The ζ-p was measured at 37 °C in mQ water as a medium, and an applied voltage of 100 V was used. The P5 sample was loaded into pre-rinsed folded capillary cells, and twelve measurements were performed. 

### 2.7. Microbiology

#### 2.7.1. Microorganisms

A total of 61 isolates belonging to several Gram-positive and Gram-negative species were used in this study. All were clinical strains isolated from human specimens and identified by VITEK^®^ 2 (Biomerieux, Firenze, Italy) or matrix-assisted laser desorption/ionization time-of-flight (MALDI-TOF) mass spectrometric technique (Biomerieux, Firenze, Italy). Of the tested 23 Gram-positive organisms, ten strains belonged to the *Enterococcus* genus, (four *Enterococcus faecalis* resistant to vancomycin (VRE), three *E. faecium* VRE, one *E. casseliflavus* (intrinsecally resistant to vancomycin), one *E. durans* and one *E. gallinarum* (intrinsecally resistant to vancomycin), 12 strains pertained to the *Staphylococcus* genus, including two methicillin resistant *S. auresus* (MRSA) and one susceptible, three methicillin resistant *S. epidermidis* (MRSE) two of which were also resistant to linezolid, one *S. haemolyticus* methicillin resistant (MR), one *S. hominis* MR, one *S. lugdunensis*, one *S. sapropyticus*, one *S. simulans* MR and one *S. warneri.* A strain of the sporogenic *Bacillus subtilis* was also added. Reguarding the thirty eight Gram-negative isolates, 18 strains were *Enterobacteriaceae*: 3 *Escherichia coli* (one was susceptibile to all antibiotics tested and one was a O157:H7 strain), one *Proteus mirabilis*, one *Morganella morganii*, one *Providencia stuartii*, one group B *Salmonella*, one *Serratia marcescens*, one *Yersinia enterocolitica*, six group A carbapenemase-producing *Klebsiella pneumoniae*, two non-carbapenemase-producing *K. pneumoniae* and one *K. oxytoca.* Twenty strains belonged to the non-fermenting group: six *Pseudomonas aeruginosa*, one *P. fluorescens*, one *P. putida*, six *Stenotrophomonas maltophylia*, five *Acinetobacter baumannii* and one *A. pittii*.

#### 2.7.2. Determination of the MIC and MBC

To investigate the antimicrobial activity of M5 and P5 on the 61 pathogens, their Minimal Inhibitory Concentrations (MICs) were determined by following the microdilution procedures, detailed by the European Committee on Antimicrobial Susceptibility Testing (EUCAST) [16]. 

Briefly, overnight cultures of bacteria were diluted to yield a standardized inoculum of 1.5 × 10^8^ CFU/mL. Aliquots of each suspension were added to 96-well microplates containing the same volumes of serial 2-fold dilutions (ranging from 1 to 512 μg/mL) of M5 or P5 to yield a final concentration of about 5 × 10^5^ cells/mL. The plates were then incubated at 37 °C. After 24 h of incubation at 37 °C, the lowest concentration of M5 or P5 that prevented a visible growth was recorded as the MIC. All MICs were obtained in triplicate, the degree of concordance in all the experiments was 3/3 and THE standard deviation (±SD) was zero. The Minimal Bactericidal Concentration (MBC) has been defined as the lowest concentration of a drug that results in killing 99.9% of the bacteria being tested [17]. 

The MBCs of M5 and P5 on the 61 pathogens were determined by subculturing the broths used for MIC determination. A quantity of 10 μL of the culture broths of the wells, corresponding to the MIC and to the higher MIC concentrations, were plated onto fresh MH agar plates, and further incubated at 37 °C overnight. 

The highest dilution that yielded no bacterial growth on the agar plates was taken as the MBC. All tests were performed in triplicate and the results were expressed as the mode.

#### 2.7.3. Killing Curves

Killing curve assays for P5 were performed on three representative isolates of *P. aeruginosa* (strains 247, 256, and 259), two representative strains of *K. pneumoniae* (strains 366 and 369, both producing class A carbapenemases) and two representative isolates of *S. aureus* (strain 18 and 195, both MRSA), as previously reported [18]. Experiments were performed over 24 h at P5 concentrations of four times the MIC for all strains.

A mid logarithmic phase culture was diluted in Mueller–Hinton (MH) broth (Merck, Darmstadt, Germany) (10 mL) containing 4 × MIC of the selected compound to give a final inoculum of 1.0 × 10^5^ CFU/mL. The same inoculum was added to cation-supplemented Mueller–Hinton broth (CSMHB) (Merck, Darmstadt, Germany), as a growth control. Tubes were incubated at 37 °C with constant shaking for 24 h. Samples of 0.20 mL from each tube were removed at 0, 30 min, 2, 4, 8, and 24 h, diluted appropriately with a 0.9% sodium chloride solution to avoid carryover of P5 being tested, plated onto MH plates, and incubated for 24 h at 37 °C. Growth controls were run in parallel. The percentage of surviving bacterial cells was determined for each sampling time by comparing colony counts with those of standard dilutions of the growth control. The results have been expressed as log_10_ of viable cell numbers (CFU/mL) of surviving bacterial cells over a 24 h period. Bactericidal effect was defined as a 3 log_10_ decrease of CFU/mL (99.9% killing) of the initial inoculum. All time-kill curve experiments were performed in triplicate.

#### 2.7.4. Evaluation of the Antimicrobial Effect of P5 by Turbidimetric Studies

The study of the antimicrobial activity of P5 was carried out measuring the optical density variations (OD) as a function of time in cultures of the same strains employed for the time-killing experiments (three trains of *P. aeruginosa*, two of *K. pneumoniae*, and two of *S. aureus*) at a wavelength of 600 nm in a Thermospectronic spectrophotometer (Ultrospec 2100pro, Amersham Biosciences, Little Chalfont, UK) [19].

Bacterial cells were harvested from 10 mL of bacterial cultures in MH broth, and cell number was adjusted to produce a heavy inoculum (OD adjusted to 0.2) corresponding to 10^8^ cells/mL. Cell suspensions were treated with or without P5 at concentration equal to 4 MIC and incubated at 37 °C. After 30 min and 1, 2, 3, 4, 5, and 6 h of incubation, aliquots were taken from the cultures, and absorbance values were recorded at 600 nm. Measurements were blanked with MH broth containing an equivalent amount of P5 being tested. The experiments were performed in triplicate. The number of CFU was determined in parallel, as described in the time-killing section and compared with the untreated sample.

## 3. Results and Discussion

### 3.1. Synthesis and Spectrophotometric Characterization of 4-(4-Amminobuthyl)styrene Hydrochloride M5 (***5***)

Monomer M5 (**5**), whose structure was designed on the base of structure/activity relationship as reported in Section 3.6.1, was synthesized according to Scheme 1, starting from commercially available 4-chlorovinylbenzene to obtain a lab-prepared Grignard reagent (**1**), subsequently exploited to prepare the brome buthyl styrene derivative (**2**), applying a modified version of a procedure proposed by Bertini et al. (2004) [12]. 

Briefly, **2** was achieved through the copper halide-coupling reaction of 1,4-dibromobuthane with **1** in the presence of LiCuBr_2_ as a catalyst [14,20]. To limit the production of the product derived from the side reaction of double alkylation, the reaction was carried out in a strong excess of dibromide, which was removed largely by high vacuum distillation to prevent spontaneous polymerizations. The isolation of **2** was then preferentially performed by column chromatography which allowed to isolate the double alkylated compound by using a mixture petroleum ether/acetone 1/1. The successive reaction of **2** with potassium phthalimide in DMF afforded the Gabriel adduct (**3**), which was purified and characterized before submission to hydrazinolysis by treatment with hydrazine in EtOH 95 °C, with heating. The obtained 4-aminobuthystyrene (**4**) was promptly transformed into the corresponding hydrochloride (**5**), which was easier to be purified and stored. Anyway, since highly hydroscopic, M5 was stored in a drier under vacuum. 

The fourier-transform infrared (FTIR) spectroscopy spectrum of **5** showed the typical vinyl double bond bands at 911 and 991 cm^−1^, two intense bands at 2930 and 2978 cm^−1^ relating to the absorptions of the aliphatic chain methylene groups and that of the protonated amino group at 3435 cm^−1^ (Figure 1).

In addition to the inevitable picks at 3.35 and 4.90 ppm, belonging to the deuteromethanol (CD_3_OD) used as solvent for the acquisition, the ^1^H NMR spectrum of M5 showed a multiplet signal centered around 1.71 ppm (-CH_2_CH_2_-), a triplet at 2.68 ppm (CH_2_-phenyl) and a triplet at 2.94 ppm (CH_2_NH_3_^+^). Although this latter group should give a more complex multiplet, a simple triplet was observed, due to the unique couplings of the CH_2_ group with the protons of the adjacent methylene of the alkyl chain and the absence of couplings with the protons of the NH_3_^+^ group, as these undergo exchange in the working solvent. The vinyl system provided the typical signal, consisting of 2 double doublets, that are made up of integrals denoted one proton for doublet (5.17 and 5.72 ppm), associated with a quartet at 6.71 ppm. The *p*-di-substituted aromatic system provided a multiplet centered at 7.27 ppm, where the integral denoted 4 protons (Figure 2a).

The ^13^C NMR spectrum presented 10 signals, plus the multiplet signal close to 50 ppm, belonging to CD_3_OD used as solvent for acquiring the spectrum. 4 signals for the carbons of the aliphatic chain, being the signal at 40.70 ppm that of methylene bound to nitrogen, were observed. Two signals for the vinyl system (113.27 and 129.71 ppm), two intense signals related to the 4 aromatic carbon atoms (127.32 and 129.71 ppm) and two low-intensity signals relating to quaternary aromatic carbons (136.94 and 142.65) were also detectable (Figure 2b).

### 3.2. Preparation of Copolymer P5 by Radical Copolymerization in Solution and Its Spectroscopic Characterization

Cationic random copolymers are the most reported macromolecular architectures studied as antimicrobial agents. They can be easily synthesized by merging an uncharged comonomer with a cationic monomer, thus, obtaining structures having charged, and not charged, moieties ‘randomly’ isolated along the polymer backbone [8]. Utilizing the more suitable comonomer and adjusting feed ratios, the amphiphilic character, and the hydrophobic content, which are strongly correlated with the antimicrobial activity and selectivity of copolymers, can be tunable [8]. In this context, preliminary studies of radical polymerization had showed that monomer M5 (**5**) omopolymerized and copolymerized easily with different comonomers. Among them, and in sight of a possible clinical utilization of our copolymer, di-methyl acrylamide (DMAA) was considered as an optimal hydrophilic comonomer, capable to provide water-soluble macromolecules, easily administrable in vivo [12,21]. Couples solvent/initiator as water/ammonium-persulfate (APS) and methanol or DMF/AIBN afforded conversions in the range 20–94% [12]. In the present work, the copolymerization of **5** with DMAA was performed in DMF/AIBN at 60 °C achieving the random copolymer (P5) with a conversion of 46% (Scheme 2). 

The experimental data of the copolymerization have been reported in Table 1.

The copolymer P5 was purified by repeated cycles of dissolution/precipitation using MeOH as solvent and Et_2_O as non-solvent. In addition, a sample of P5 was subjected to fractioning. Fractioning was performed at 25 °C, adding an excess of Et_2_O to the subsequently isolated oily precipitates of P5, achieving high Mn and low Mn co-polymers of P5 free from traces of monomer and co-monomer, as confirmed by the NMR spectra lacking the typical signals of the double doublets of the vinyl system.

In the FTIR spectra of P5 (Figure 3) an intense broad absorption around 3000–3500 cm^−1^ due to NH_3_^+^ groups deriving from **5** was present (not showed), confirming the contribute of M5. In addition, the typical single band of phenyl ring *o*-di-substituted between 800 and 750 cm^−1^ belonging to M5 and the amide band of DMAA around 1650 cm^−1^ were detected. 

The copolymer was soluble in water, methanol, DMSO and DMF, while insoluble in petroleum ether, diethyl ether and acetone. P5 was partially soluble in THF and swelled in cold toluene and dioxane.

### 3.3. Determination of the Average Molecular Mass (Mn) of Copolymer P5

#### 3.3.1. The Technique

The average molecular mass (Mn) of P5 has been determined by using the vapor-pressure osmometry (VPO) method in MeOH at 45 °C. VPO is an experimental technique for the determination of a polymer’s number average molecular mass, Mn. It works by taking advantage of the decrease in vapor pressure that occurs when solutes are added to pure solvent. This technique can be used for polymers with a molecular weight of up to 20,000 though accuracy is best for those below 10,000 [22].

Higher polymers can be analyzed using other techniques such as membrane osmometry and light scattering. Nowadays, VPO faces competition from matrix-assisted laser desorption ionization mass spectrometry (MALDI-MS), but VPO still has some advantages when fragmentation of samples for mass spectrometry may be problematic [23].

A typical vapor phase osmometer consists of two thermistors, one with a polymer-solvent solution droplet adhered to it and another with a pure solvent droplet adhered to it. Several syringes are employed to provide thermistors with pure solvent or solution drops. The thermistors are in a cell where the gas phase is saturated with solvent vapor. The cell temperature is electronically controlled and maintained with an accuracy of ±1 × 10^−3^ °C. The operating temperature can be selected in the range between 20 and 130 °C. An electric circuit measures the bridge output imbalance difference between the two thermistors. The increasing vapor pressure of the solution droplet leads to an increase of temperature. Once equilibrium is reached, a constant measurement value is achieved. This ∆T between the thermistors is always proportional to the number of particles or number of moles dissolved in the solution. Consequently, when the sample concentration is known, average molecular mass (Mn) can be determined. 

#### 3.3.2. Calibration and Measurements

By using PEO 10800 as selected standard for our determinations, in the first phase of calibration a linear regression curve was developed, and its data have been reported in Table 2, and which allowed to calculate K_cal_, corresponding to the value on the y axis when c (mol/Kg) is zero. 

In the second phase, measurements were executed on methanol solutions of P5 whose concentrations (g/Kg) have been reported in Table 2 and the *K_meas_* (Kg/g), corresponding to the value on the y axis when c (g/Kg) is zero, was provided by the instrument (Table 2). *K_meas_* was used to estimate the Mn of P5 according to Equation (2) reported in Section 2.4.2.

### 3.4. Determinations of NH_2_ Content of P5

To determine the NH_2_ content of P5 and have evidence of its density of charge the titration of amine hydrochlorides with HClO_4_ solution in acetic acid (AcOH) in the presence of mercuric acetate and quinaldine red as indicator [15] proved to be simple and reliable. The method is cheap and fast, and its accuracy has been secured by a sharp endpoint of titration, while its reliability has been demonstrated by the reproducibility of results (Table 3). 

### 3.5. Particle Size, ζ-p and PDI of P5

The hydrodynamic size (diameter) (Z-AVE, nm) and PDI of P5 were determined by DLS analysis, and ζ-p measurements were carried out to determine the surface charge of P5. The results have been reported in Table 3.

P5 particles showed an average size of 334 nm with a PDI of 1.012, which denotes a significant polydispersity with presence of particles with variable size. Although a high PDI could mean low stability of polymer solution due to a possible tendency forming aggregates, P5 water solutions proved to be stable and to remain clear along time both at room temperature and under heating. The ζ-p (+58 mV) was significantly higher than the value of 30 mV, which is considered a critical value below which, low stability in solution and a tendency to form aggregates can occur. As expected ζ-p of P5 was positive, confirming its cationic charge due to the presence of ammonium groups in water solution. Furthermore, ζ-p of P5 was in accordance with those of cationic polystyrene-based nanoparticles recently prepared by co-polymerizing styrene with the *N*-(2-(methacryloyloxy)ethyl)-*N*,*N*-dimethyltetradecane-1-ammonium bromide (MDTP) as active monomer [24]. Compound named CNPS-4, having the highest content of cationic monomer (80%), but a lower Mn if compared to P5, displayed and identical value of ζ-p (+58 mV).

Polymer particles with high positive ζ-p usually are capable of faster absorption on cells surface by electrostatic interactions, with an eventual easier internalization than particles with negative ζ-p. Consequently, the high ζ-p of +58 mV, observed for P5 could decree for a possible activity as membrane disruptor and for a broad-spectrum antibacterial effect simply on contact with bacterial wall.

### 3.6. Antibacterial Properties

#### 3.6.1. Design of the Structure of the Cationic Monomer M5 and the Choice of the Not Charged Comonomer (DMAA)

Macromolecules deriving from polymerization of aromatic monomers containing quaternary nitrogen atoms, such as *N*-alkylated 4-vinylpyridine, have been reported to kill up to 99% of drug-resistant Gram-positive, Gram-negative bacteria and fungi [25,26,27]. To address the issue of their minimal biocompatibility, copolymers of *N*-alkylated 4-vinylpyridine with not cationic co-monomers as diluents, such as hydroxyethyl methacrylate and poly(ethylene glycol) methyl ether methacrylate, were prepared to attenuate the cationic charge density responsible for cytotoxicity [27]. As a result, antibacterial copolymers with enhanced bactericidal activity due to higher wettability, and improved biocompatibility were achieved [26]. 

In this scenario, in order not to use the already extensively studied 4-vinylpyridine monomer commercially available, and to avoid the post-polymerization reactions to convert the nitrogen atom into a quaternary salt, as already described, we designed our monomer as a styrene derivative, rather than a pyridine one.

In this regard, we inserted in *para* position the alkyl chain possessing a primary amine group in the form of hydrochloride salt as cationic group. 

The primary ammonium group was preferred to the permanently protonated quaternary ammonium groups. This is because among the various studies reported, several co-polymers were synthetized containing alkyl chains terminated by primary amine groups in the form of acid salts, which proved to possess considerable broad-spectrum antimicrobial properties, often higher than those of quaternary ammonium analogous [2]. 

Accordingly to what reported, the C6 alkyl chain proved to be the best solution to obtain the highest killing efficacy, followed by the C3 and C4 chains. Longer chains as the C8-C16 ones were significantly less effective [26,27]. Unfortunately, attempts to co-polymerize a monomer containing the C6 chain as linker between the aromatic ring of styrene and the ammonium group afforded material with inadequate physicochemical properties. We recovered to the insertion of a C4 chain preparing M5, which was employed in the reaction of copolymerization, using DMMA as hydrophilic comonomer for assuring high wettability and for the reasons already explained (Section 3.2).

#### 3.6.2. Antimicrobial Activity of P5

MIC values for P5 were obtained analyzing a total of 61 strains of clinical origin including both Gram-positive and Gram-negative species. For comparison, the monomer M5 (**5**) was analyzed under the same conditions. Although against *S. aureus* it displayed MICs considerably lower than those of previously reported monomers [28], according to other case studies [2,3,28,29,30,31], we considered M5 ineffective against all isolates in this study. On the contrary, the macromolecular compound P5 provided remarkably interesting results both against Gram-positive and Gram-negative species (Table 4).

Extremely low MIC values were observed against all representatives of genus *Enteroccoccus* (VRE included) and *B. subtilis* (MIC = 3.15–6.3 µM), with MBC values only twice the MIC, with the exception of *E. casseliflavus* 184, against which the MBC value was superimposable to the MIC. Concerning *Staphylococci*, similar MIC values were observed against *S. epidermidis* (MIC = 6.3 µM) and other minor *Staphylococci* (MIC values = 3.15–6.3 µM), while MIC values slightly higher were measured against *S. aureus* (MIC = 12.6 µM). However, P5 proved to be more potent, against MRSA (MIC of PAI2 = 14.9 µM), than the best performant macromolecule (PAI2) belonging to a family of three cationic random copolymers (PAI1-PAI3), containing alkyl ammonium hydrochloride moieties and aromatic rings as P5 [32]. In addition, P5 was extremely more potent than PAI1 and PAI3, having Mn like that of P5. Indeed, a MIC = 17.7 µM was observed for PAI3 (Mn = 5300) and a MIC = 267.8 µM was displayed by PAI1 (Mn = 5600). Against *S. epidermidis*, P5 was slightly less effective than PAI2 and PAI3, but more potent than PAI1. Moreover, P5 was more active than a primary amine-containing polystyrene-based polymer (Poly**1**) previously reported as endowed with antibacterial activity comparable to that of a potent derivative of the peptide magainin II, and with an antibacterial profile like that of the peptide toxin melittin. It displayed MIC values 1.6-fold lower against both MRSA and *E. faecium* VRE strains [3]. With regard to Gram-negative species, P5 displayed low MIC values against different strains of *K. pneumoniae* including carbapenemase (KPC)-producing isolates (MICs = 6.3–12.6 µM), four isolates of *A. baumannii* and *A. pitti* (MICs = 6.3 µM), several KPC-producing *S. maltophilia*, a *Salmonella gr.* B isolate (MICs = 3.15–12.6 µM), *P. aeruginosa*, *P. putida* and *E. coli* (MIC = 6.3 µM), and an even lower MIC value was observed against *P. fluorescens* (MIC = 3.15 µM). In all cases, the corresponding MBC values were the same of the MICs or twice the MICs. As expected, P5 proved to be ineffective against *P. aeruginosa* resistant to colistin (result not included in Table 4). Indeed, as reported in the literature [33,34], polymyxin resistance in *P. aeruginosa*, as well as in other Gram-negative bacteria, is associated with the addition of 4-amino-*L*-arabinose (L-Ara4N) or phosphoethanolamine (pEtN) to lipid A and *core* oligosaccharide components. This process causes a decrease in the net negative charge of the outer membrane, and in turn, the reduction of the possible interactions drug/bacteria necessary for the antibacterial effect. For similar reasons, very high MIC values were observed against *Serratia marcescens* (MIC > 100 µM), *Morganella morganii* (MIC = 50 µM) and *P. mirabilis* 254 (MIC = 25 µM), which as reported, can develop resistance against cationic agents like CAMPs substituting the lipid A with aminoarabinose [34,35]. In relation to *Y. enterocolitica*, the high MIC value observed can be justified by the capability of these pathogens to increase resistance versus CAMPs, including polymyxin B, cecropin P1 and melittin, as well as against cationic compounds acting as CAMPs, due to unspecified LPS modifications, possibly mediated by an improved expression of RosAB efflux pump system [36]. The MIC values observed against *Providencia stuartii*, although appearing high, were comparable or even lower than those reported for cationic antimicrobial peptides such as Pexiganan (6.5–104.2 µM) and Citropin 1.1 (39.6–158.5 µM) [37]. 

According to previously reported results, MIC values of P5 against *E. coli* were 1.6-fold lower than that observed for Poly**1** [3]. Furthermore, MIC values of P5 against *E. coli* were from 2.4 to 3.4 times lower than those of two non-hemolytic self-degradable antimicrobial copolymers [P9 (MIC = 63 µM) and P7 (MIC = 21.7 µM)], which contained cationic side chains bearing primary ammonium salts like P5 and have Mn comparable to that of P5 [38]. In addition, recently, Wen and co-workers (2019) reported the synthesis, characterization, and antibacterial properties of four polystyrene-based cationic nanoparticles (CNPS-1-4), which are different each other in their content in quaternary cationic monomer (MDTP), molecular weight (Mn), particle size and density of charge, in terms of ζ-p, which increased with the increasing of MDTP content [24]. The study proved that the antibacterial activity of CNPSs increased with increasing MDTB content in formulae, and the minimum values of MIC were observed for CNPS-4, containing 80% MDTP and having ζ-p of +58 mV (MIC = 48.8 µM against *E. coli* and 25.0 µM against *S. aureus*).

In accordance with previous reports [1,2], these findings confirmed that increasing cationic groups in macromolecular formulae, and consequently increasing density of positive charges on the surface of polymers, enhance the interaction between cationic (co)polymers and the negatively charged bacteria, thereby leading to a higher disturbance in their physiological function, growth inhibition and death [24]. Although P5 contained much less cationic monomer (30% by moles), than CNPS-4, it showed the same identical ζ-p and charge density, due to its higher molecular weight.

Despite the density of charge of our co-polymer was equal to that of CNPS-4, P5 showed to be much more active against both *E. coli* (6.3 µM) and *S. aureus* (12.6 µM), thus, establishing that the strategy in adopting primary amine groups, in the form of hydrochloride salts in place of the extensively used quaternary ammonium group is successful.

Very promising MIC values were observed against several clinically isolates of *K. pneumonia* (MIC = 6.3 µM), a powerful superbug and hypervirulent form of the *Klebsiella* species, which can cause a variety of infections, including rare but life-threatening liver, respiratory tract, bloodstream, and other conditions [39]. 

In a recent study, Weiyang et al. (2018) [39] reported the considerable antibacterial activity of two types of polyionenes against 20 clinical strains of K. pneumoniae responsible for lung infections, which is also associated with more rapid killing kinetics than imipenem and other commonly used antibiotics.

Considering that the MICs displayed by the two polymers namely 2a and 2b were in the range 1.5–40 µM, in three cases, P5 proved to be more active by 1.6, 3.2 and 6.4 times. Whereas, in two cases, it displayed comparable activity on only in two case less active by 2 and 1 time [39]. 

#### 3.6.3. Time-Killing Curves

Time kill experiments were performed with P5 at concentrations equal to 4 × MIC on three strains of *P. aeruginosa*, two strains of *K. pneumoniae*, and two of *S. aureus*. As depicted in Figure 4, showing the most representative curves obtained for each species, P5 possessed an extremely strong bactericidal effect against all the pathogens tested, since a rapid decrease of >4 logs in the original cell number was evident already after 30 min of exposure and was maintained for 24 h after incubation.

Regrowth was noted after 24 h of incubation with P5 for all the three species tested. Interestingly, this behavior is positively different from that already observed for cationic bactericidal peptides, as colistin [40], dendrimers [5] and polymers, that kill on contact, where the initial killing is rapid, being produced as soon as 5 min [40], 1 h [5] and 1–4 h [41], respectively, after antibiotic exposure, but is followed by regrowth after 24 h.

#### 3.6.4. Effect of P5 on the Growth Curve of *P. aeruginosa*, *K. pneumoniae*, and *S. aureus*

The kinetics of growth in MH broth in the absence or presence of P5 at a concentration of 4 × MIC was followed at 600 nm for a period of 6 h on selected strains of *P. aeruginosa*, *K. pneumoniae*, and *S. aureus*. Figure 5 shows the results obtained on one representative strain of *P. aeruginosa*, *K. pneumoniae* and *S. aureus*. While, as expected, the control culture showed an exponential turbidimetric increase, the presence of P5 resulted in a complete inhibition of growth not followed, over the whole period of the experiments, by a decrease of the optical density. This suggests that despite the compound being highly bactericidal, as confirmed in the previously reported experiments, it is unable to induce frank cell lysis. Similar results were obtained for all the selected strains tested and were analogous to those obtained in our previous study where the antibacterial activity of cationic dendrimers is reported [5].

Concerning the mechanism of action of P5, it is possible to advance the hypothesis that it could perform like other CAPs. An electrostatic attraction to the highly anionic outer membrane (OM) of Gram-negative strains or to the cell wall of Gram-positive isolates of all the susceptible species, could be assumed. Thereafter, the displacement of Ca^+^ and Mg^+^ ions, destabilization of LPS, pores formation, diffusion towards CM and further electrostatic absorption, followed by further pores formation, CM destabilization, diffusion in the cytoplasm and cell death, could occur in the case of Gram-negative strains. Similarly, for Gram-positive bacteria, the electrostatic interaction of P5 with cell wall and the consequent permeabilization and pores formation could favor the diffusion of P5 towards CM, its permeabilization and damage, diffusion into cytoplasm, leading to cell death, as well. Both Gram-negative and Gram-positive strains can die because of several events, such as loss of membrane potential, inhibition of biosynthetic pathways involving ATP, DNA, RNA or proteins, free radical production, and leakage of the cytoplasmic content including crucial inorganic species, such as K^+^, Cs^+^, Na^+^, Li^+^, and phosphate [1,2,42]. The inactivity of P5 on the colistin-resistant *P. aeruginosa* strain included in our study and on other isolates which possess non-conventional membranes with inverted charge as *Y. enterocolitica* 342, *P. stuartii* 374, *M. morganii* 372, *S. marcescens* 228, and *P. mirabilis* confirms the above-mentioned hypotheses. However, it should be remembered, that the lytic mechanism attributed to the currently utilized membrane-damaging antimicrobials, such as colistin. Although, still controversial, resides in the presence of an *N*-terminated hydrophobic fatty acid side chain. This configuration, added to the positively charged peptide ring, confers to the molecule the amphiphilic character necessary for allowing its diffusion through the OM and towards the CM (Gram-negative) or through CM and towards cytoplasm (Gram-positive), pivotal for the cell lysis [2,40]. The absence of a similar fatty acid side chain in P5, by enhancing its hydrophilic character, may limit its diffusion and CM lysis. Therefore, justifying the fact that, while being strongly bactericidal, it induces irremediable and lethal membrane impairments, such as depolarization and destabilization, it lacks lytic properties [2].

## 4. Conclusions

In line with the scope of this study, we have prepared a new water-soluble cationic random copolymer, containing 4-ammoniumbuthylstyrene hydrochloride moieties (P5), which proved to have nanosized particles of 334 nm, ζ-p value of +58 mV and PDI of 1.2, and which was shown to be stable in solution without a tendency to form aggregates. 

The antibacterial and bactericidal properties and the possible mechanism of the biocidal activity of P5 were investigated by determining the MIC and the MBC values, by performing time killing and turbidimetric experiments and by determining the variations of the values of Log_10_(CFU/mL), and of the optical density at 600 nm (OD 600 nm), respectively. The better values of MIC observed against the main families of tested Gram-positive and Gram-negative species have been summarized in Figure 6a. While the differences in the values of Log_10_(CFU/mL) and OD 600 nm, observed against MDR *P. aeruginosa*, *S. aureus* and *K. pneumonia* isolates, in the absence and in presence of P5, have been reported in Figure 6b.

Collectively in this study, a new potent bactericidal molecule, capable of killing several strains of different species and belonging to both Gram-positive and Gram-negative bacteria, has been developed (P5). Sensitive bacteria were eradicated regardless of their antibiotic resistance. We conclude that, due to its physicochemical properties and water solubility, P5 could represent a novel agent capable of counteracting infections sustained by several pathogens resistant to the presently available antibiotics. 

## Data Availability

All necessary data are already included in the present study.

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
