# Peer review of "Synthesis, Characterization, and Bactericidal Activity of a 4-Ammoniumbuthylstyrene-Based Random Copolymer"

_polymers, 2021, doi:10.3390/polym13071140_

Round 1

Reviewer 1 Report

In this work, the authors reported the synthesis and physicochemical characterization of a water-soluble cationic copolymer (P5). Especially the antibacterial activity of P5 was evaluated by the test of bactericidal activity. This manuscript was organized well but its English needs to be improved. With some new results, this work may be of interest to many researchers in this area. The manuscript can be considered for publication after major revisions, considering the following questions:

  • In the section of 2.2 , NaCN should be a useless additive in the reaction system, which probably affects the reaction.
  • In Scheme 1, the structure of 4-chlorostyrene after the Grignard reaction, should be changed in the written molecular formula with “Mg” and others, because the Grignard agent was used. Please revised it.
  • Based on the 1H NMR spectrum of monomer M5 in Figure 2, it indicates the presence of impurity in the product. The authors should explain it.
  • The antibacterial activity of the product copolymer should be compared with some previous materials to show the advantages of the product.

Author Response

In this work, the authors reported the synthesis and physicochemical characterization of a water-soluble cationic copolymer (P5). Especially the antibacterial activity of P5 was evaluated by the test of bactericidal activity. This manuscript was organized well but its English needs to be improved. With some new results, this work may be of interest to many researchers in this area. The manuscript can be considered for publication after major revisions, considering the following questions:

We thank the Reviewer for his general comments, which, in addition to a useful suggestion concerning the English, also contain positive opinions on our manuscript.

Following the suggestion of the Reviewer, the manuscript has been revised by a native English expert, whose contribute has enabled us to significantly improve the quality of our work.

In the section of 2.2, NaCN should be a useless additive in the reaction system, which probably affects the reaction.

Concerning this comment by the Reviewer, we kindly make him know that we reproduced a reconized and optimized procedure previously reported in more than one published articles, where the hydrolitic system adopted is exactly that reported by us, i.e. an aqueous solution of weithed amounts of NH4+Cl- and NaCN (or KCN). Please, consider the following descriptions extracted by two articles reporting the synthesis of several bromo-alkyl-styrene derivatives.

  • From Andringa, H.; Hanekamp, ;  Brandsma,  L. Synth. Commun. 1990, 20, 2349–2351 (Ref 20 in the manuscript):

The solution was then cooled to room temperature and subsequently hydrolysed with a solution of 30 g of ammonium chloride and 5 g of potassium cyanide in 200 ml of ice water (vigorous shaking)”.

  • From Bertini, V.; Alfei, S.; Pocci, M.; Lucchesini, F.; Picci, N.; Iemma, F. Monomers containing substrate or inhibitor residues for copper amine oxidases and their hydrophilic beaded resins designed for enzyme interaction studies. Tetrahedron 2004, 60, 11407–11414 (Ref. 12 in our manuscript):

4.3. 4-(u-Bromoalkyl)styrenes 3c and 3d. A mixture of thea,u-dibromoalkane (50 mmol), dry THF(20 mL) and a solution of LiCuBr (1.5 mL) in dry THF was cooled   to   0°C,   treated   dropwise   with   0.68 M 4-vinylphenyl magnesium chloride (18 mL, 12.2 mmol) in THF and stirred at room temperature for 5 h. The reaction mixture was then treated with an iced aqueous solution of NaCN (0.80 g) and NH4Cl (5.00 g) dissolved in water (35 mL) and extracted with peroxide-free ethyl ether (4x30 mL)”.  

In addition, considering the outcomes of the reaction (77% yield after double purification, i.e. distillation and chromatographic column), despite the inevitable formation of the dialkylation side-product, which negatively affects the final yield, it is possible to assert that there is nothing wrong with the synthetic and hydrolysis procedure adopted by us and, before us, by others.

Indeed, since in this reaction the final hydrolysis mainly serves to destroy the residual Grignard used in excess, the use of the hydrolytic system containing NaCN and NH4Cl operates to make the hydrolysis gentler. In fact, NaCN by reacting with NH4Cl, produces NH4CN, which being instable, decomposes to ammonia and HCN which is a weakly acid (pKa = 9.2) if compared to HCl, which would derive by the decomposition of NH4Cl, if used alone.

In Scheme 1, the structure of 4-chlorostyrene after the Grignard reaction, should be changed in the written molecular formula with “Mg” and others, because the Grignard agent was used. Please revised it.

We apologise with the Reviewer for our distraction in reproducing the structure of compound 1 in which the magnesium atom is missing. The Scheme 1 has been corrected and a new image has been inserted. Please, see page 10.

Based on the 1H NMR spectrum of monomer M5 in Figure 2, it indicates the presence of impurity in the product. The authors should explain it.

We thank the Reviewer for his comment, which gives us the possibility to improve our manuscript, with a more clear explanation concerning the NMR spectra of M5. As already explained in the main text of the original manuscript, both 1H and 13C NMR spectra of M5 present signals in accordance with its structure and in particular, in the 1H NMR spectrum, also the the multiplicity of signals was found to be correct. In this regard, additional explanations concerning the multiplicity of the signals have been inserted (lines 414-418). However, as the Reviewer points out, there are more signals than those explained. These signals belong to the solvent used to acquire the spectra. In fact, we explain, that, despite deuterated solvents are used, and in the present case deuteromethanol (CD3OD), there is never a deuteration of 100%, so that the residual hydrogen atoms inevitably give peaks to precise and tabulated ppm values, both in the 1H and 13C NMR. In particular, the CD3OD gives two peaks in the proton spectrum, a doublet around 3.35 ppm and a multiplet around 4.9 ppm. As far as carbon is concerned, the CD3OD gives a signal in the form of a multiplet, just below 50 ppm. Based on these considerations, the spectra of compound M5 do not show any impurities, but the peaks of the substance plus those of the solvent. We incorrectly thought, that putting the diciture MeOH close to the peaks not belonging to M5 might be enough, but now, we realize that it is not. Therefore, the Figure 2a and 2b, have been modified, by inserting more details to make the question clearer. In addition some lines have also been added in the text. Please, see lines 411-412, 424-425 and 427-428.

The antibacterial activity of the product copolymer should be compared with some previous materials to show the advantages of the product.

We make kindly note to the Reviewer, that several comparisons between the activity of P5 on several bacterial species of those reported in our manuscript, and that of previous materials, were already present in the non revised, original manuscript. Please consider lines 594-602, following reported:

However, P5 proved to be more potent, against MRSA (MIC of PAI2 = 14.9 µM), than the best performant macromolecule (PAI2) belonging to a family of three cationic random copolymers (PAI1-PAI3), containing alkyl ammonium hydrochloride moieties and aromatic rings as P5 [32]. In addition, P5 was extremely more potent than PAI1 and PAI3, having Mn similar to that of P5. Indeed, a MIC = 17.7 µM was observed for PAI3 (Mn = 5300) and a MIC = 267.8 µM was displayed by PAI1 (Mn = 5600). Against S. epidermidis, P5 was slightly less effective than PAI2 and PAI3, but more potent than PAI1. Moreover, P5 was more active than a primary amine-containing polystyrene-based polymer (Poly1) previously reported as endowed with antibacterial activity comparable to that of a potent derivative of the peptide magainin II, and with an antibacterial profile like that of the peptide toxin melittin. It displayed in fact MIC values 1.6-fold lower against both MRSA and E. faecium VRE strains [3].

Moreover, consider also lines 623-626 following reported:

The MIC values observed against Providencia stuartii, although appearing high, were comparable or even lower than those reported for cationic antimicrobial peptides such as Pexiganan (6.5-104.2 µM) and Citropin 1.1 (39.6-158.5 µM) [37]. 

And furthermore, please, consider lines 636-672:

According to previously reported results [3,38], the MIC value of P5 against E. coli was 1.6-fold lower than the MIC value observed for Poly1 [3]. Furthermore, it was from 2.4 to 3.4 times lower than those of two not hemolytic self-degradable antimicrobial copolymers (P9 and P7) bearing similar cationic side chains in the form of primary ammonium salts and having comparable Mn [38]. In addition, recently, Wen and co-workers (2019) reported the synthesis, characterization and antibacterial properties of four polystyrene-based cationic nanoparticles (CNPS-1-4), different for their content in quaternary cationic monomer (MDTP), molecular weight (Mn), particle size and density of charge, in terms of ζ-p, which increased with the increasing of MDTP [24]. The study proved that the antibacterial activity of CNPSs increased with increasing MDTB content in formulae, and the minimum values of MIC were observed for CNPS-4, containing 80% MDTP and having ζ-p of +58 mV (MIC = 48.8 µM against E. coli and 25.0 µM against S. aureus).

Accordingly with what already reported, these findings confirm that increasing cationic groups in macromolecular formulae and consequently increasing density of positive charges on the surface of polymers enhance the interaction between cationic (co)polymers and the negatively charged bacteria leading to disturbance of their physiological function, growth inhibition and death [24]. Although P5 contained much less cationic monomer (30% by moles), it showed the same identical ζ-p and charge density, due to its higher molecular weight.

Despite the density of charge of our co-polymer was equal to that of CNPS-4, P5 showed to be much more active against both E. coli (6.3 µM) and S. aureus (12.6 µM), thus establishing that the strategy of adopting primary amine groups in the form of hydrochloride salts in place of the extensively used quaternary ammonium group is successful.

Very promising MIC values were observed against several clinically isolates of K. pneumonia (MIC = 6.3 µM), a powerful superbug and hypervirulent form of the Klebsiella species, which is able to cause a variety of infections including rare but life-threatening liver, respiratory tract, bloodstream and other conditions [39].  

In a recent study, Weiyang et al. (2018) [39] reported the considerable antibacterial activity of two types of polyionenes against 20 clinical strains of K. pneumoniae responsible of lung infections, also associated with more rapid killing kinetics than imipenem and other commonly used antibiotics.

Considering that the MICs displayed by the two polymers namely 2a and 2b were in the range 1.5-40 µM, in three cases P5 proved to be more active by 1.6, 3.2 and 6.4 times, in two cases it displayed comparable activity on only in two case less active by 2 and 1 time [39].

In addition, we make kindly note to the Reviewer, that comparisons also concerning the time killing results and the turbidimetric experiments, were already present in the original manuscript.

Please consider the following parts of the original manuscript.

Lines 682-687.

No, regrowth was noted after 24 h of incubation with P5 for all the three species tested. Interestingly, this behavior is positively different from that already observed for cationic bactericidal peptides, as colistin [40], dendrimers [5] and polymers, that kill on contact, where the initial killing is rapid, being produced as soon as 5 min [40], 1h [5] and 1-4 h [41], respectively, after antibiotic exposure, but is followed by regrowth after 24 h.

Lines 720-731

However, it should be remembered, that the lytic mechanism attributed to the currently utilized membrane-damaging antimicrobials, such as colistin, although still controversial, resides in the presence of an N-terminated hydrophobic fatty acid side chain. This configuration, added to the positively charged peptide ring, confers to the molecule the amphiphilic character necessary for allowing its diffusion through the OM and towards the CM (Gram-negative) or through CM and towards cytoplasm (Gram-positive), pivotal for the cell lysis [2,40].The absence of a similar fatty acid side chain in P5, by enhancing its hydrophilic character, may limit its diffusion and CM lysis, thus justifying the fact that, while being strongly bactericidal inducing irremediable and lethal membrane impairments such as depolarization and destabilization, it lacks lytic properties [2].

We think that comparisons already present in our manuscript could be sufficient and that additional ones, would only burden the work. We therefore kindly ask the Reviewer to be satisfied.

Reviewer 2 Report

After a rather well-written introduction and a simple but clear presentation of the problem, the authors report the synthesis and physicochemical characterization of a water-soluble copolymer obtained by copolymerization of a laboratory-made monomer, 4-ammoniumbuthylstyrene hydrochloride, with dimethylacrylamide as diluent. This product was characterized by means of particle size, zeta-potential, solubility and stability in water solution. Moreover, the antibacterial activity of this compound was extensively investigated. 

The experimental work seems competently conducted and I have no particular criticism to make. The results are interesting and may be useful to investigators working in this field.

Only two small criticisms. The quality of figures 4 and 5 is very poor and should be improved. The authors should summarize their main results which are scattered here and there in the previous sections but which would be useful to summarize at the end of the manuscript.

Author Response

After a rather well-written introduction and a simple but clear presentation of the problem, the authors report the synthesis and physicochemical characterization of a water-soluble copolymer obtained by copolymerization of a laboratory-made monomer, 4-ammoniumbuthylstyrene hydrochloride, with dimethylacrylamide as diluent. This product was characterized by means of particle size, zeta-potential, solubility and stability in water solution. Moreover, the antibacterial activity of this compound was extensively investigated. 

The experimental work seems competently conducted and I have no particular criticism to make. The results are interesting and may be useful to investigators working in this field.

Only two small criticisms. The quality of figures 4 and 5 is very poor and should be improved.

We thank the Reviewer for his positive general comments on our study and we hope to furtherly satisfy him with our following resposes to his requests.

Figure 4 and 5 have been remade improving thier quality in terms of size, indicators size, thickness of the curves lines, clearness, contrast and luminosity. He hope that the work made is sufficient.

The authors should summarize their main results which are scattered here and there in the previous sections but which would be useful to summarize at the end of the manuscript.

As suggested by the Reviewer, the results obtained by evaluating P5 have been summarized at the end of the manuscript in the new Conclusions Section. In particular, concerning the antibacterial and bactericidal activity of P5 (in terms of the better MIC values displayed by P5, and its influence on the variation of Log10(CFU/mL) and of OD 600 (nm) values), have been summarized by means of two additional Figures (Figure 6 and 7). Please see lines 738-776 and the new Figures 6 and 7.